# Referable Diabetic Retinopathy Prediction Algorithm Applied to a Population of 120,389 Type 2 Diabetics over 11 Years Follow-Up

**DOI:** 10.3390/diagnostics14080833

**Published:** 2024-04-17

**Authors:** Pedro Romero-Aroca, Raquel Verges, Jordi Pascual-Fontanilles, Aida Valls, Josep Franch-Nadal, Xavier Mundet, Antonio Moreno, Josep Basora, Eugeni Garcia-Curto, Marc Baget-Bernaldiz

**Affiliations:** 1Ophthalmology Service, University Hospital Sant Joan, Pere Virgili Health Research Institute (IISPV), 43204 Reus, Spain; raquel.vergepu@gmail.com (R.V.); eugenigorg@gmail.com (E.G.-C.); mbaget@gmail.com (M.B.-B.); 2ITAKA Research Group, Department of Computer Science and Mathematics, Pere Virgili Health Research Institute (IISPV), Universitat Rovira i Virgili, 43007 Tarragona, Spain; jordi.pascual@urv.cat (J.P.-F.); antonio.moreno@urv.cat (A.M.); 3Diabetis des de l’Atenció Primária (DAP)-Cat Group, Unitat de Suport a la Recerca Barcelona, Fundació Institut Universitari per a la Recerca a l’Atenció Primària de Salut Jordi Gol i Gurina (IDIAPJGOL), 08007 Barcelona, Spain; josep.franch@gmail.com (J.F.-N.); mundetx@gmail.com (X.M.); jbasora.tarte.ics@gencat.cat (J.B.)

**Keywords:** diabetic retinopathy, algorithm, sensitivity, specificity, artificial intelligence

## Abstract

(1) Background: Although DR screening is effective, one of its most significant problems is a lack of attendance. The aim of the present study was to demonstrate the effectiveness of our algorithm in predicting the development of any type of DR and referable DR. (2) Methods: A retrospective study with an 11-year follow-up of a population of 120,389 T2DM patients was undertaken. (3) Results: Applying the results of the algorithm showed an AUC of 0.93 (95% CI, 0.92–0.94) for any DR and 0.90 (95% CI, 0.89–0.91) for referable DR. Therefore, we achieved a promising level of agreement when applying our algorithm. (4) Conclusions: The algorithm is useful for predicting which patients may develop referable forms of DR and also any type of DR. This would allow a personalized screening plan to be drawn up for each patient.

## 1. Introduction

Diabetes mellitus (DM) is a chronic disease associated with insulin resistance problems, secondary to obesity and metabolic syndrome, with its most common form being type 2 diabetes mellitus (TDM2). According to data from the International Diabetes Federation (IDF), T2DM affected 463 million people globally in 2019 and will affect 700 million people by 2045 [1]. In Spain, a study conducted by di@bet.es reported that 7.9% of the population is known to have DM, although its true prevalence is estimated to be double that, at 13.8% [2]. T2DM affects various organs of the body in the form of vascular involvement, such as stroke and myocardial infarction (more frequent conditions in patients with DM), including the kidneys and eyes, affecting the small vessels in these organs in the form of nephropathy or diabetic retinopathies. 

DR is characterized by a series of different lesions, which can be attributed to the interruption of blood supply to the retina, the breakdown of barriers, and increased vascular permeability. These changes in the retina can develop into two forms of sight-threatening diseases, proliferative diabetic retinopathy (PDR) and diabetic macular edema (DME). DR is currently described as neurovascular involvement of the retina in patients with DM 1 and 2, a concept that includes the previous description as microangiopathy or involvement of the small vessels of the retina and to which the current evidence of neuronal involvement would be added. It happens in parallel or prior to the previous one and can be detected by electroretinography [3].

DR is characterized by several different lesions, which can be attributed to the disruption of blood supply to the retina, barrier breakdown, and increased vascular permeability. These changes usually begin in the periphery of the retina and can develop into sight-threatening forms, severe diabetic retinopathy, proliferative diabetic retinopathy (PDR), and diabetic macular edema (DME). These three forms can be grouped together under sight-threatening diabetic retinopathy (STDR), which, as indicated, are forms that can lead to blindness or low vision.

Early detection of this disease aids in its diagnosis and early treatment, so screening is actively offered to the population susceptible to suffering from the disease, even those who are asymptomatic and may not have requested medical help. Recommendations to screen for DR were established long ago in the 1989 Declaration of Saint Vincent, in which it was stated that screening for DR is the cornerstone of the management and treatment of DM [4]. Digital retinography using a 45° non-mydriatic camera (CNM), due to its ease of use, is the method most commonly used in DR screening. Its association with teleophthalmology in DM allows access to screening for all diabetic people, regardless of whether they are at a disadvantage due to geography, mobility, or socioeconomic status. 

Scientific societies involved in the treatment of DM and its complications recommend annual retinal monitoring in patients with T2DM [5,6]. Despite this, compliance with these recommendations is difficult, even in developed countries [7,8]. To increase the number of patients screened, new types of technologies are currently being developed, such as automated image classification systems, hand-held cameras, and technologies based on artificial intelligence (AI) for reading and analyzing retinal images that ease the detection of DR.

Likewise, different entities have recommended extending the control period from one to two years for patients with well-managed DM, but what this looks like has not been defined. A recently published systematic review article [9] evaluated a total of 11 relevant studies, in which the interval between visits ranged between 2 and 5 years. However, only three of them used a DR prediction algorithm [10,11,12], that developed by Aspelund et al. [13]. The rest were based only on HbA1c values or individual action protocols. As far as the development of prediction algorithms using clinical data from well-controlled populations is concerned, part of Aspelund’s algorithm is currently available, as well as those developed by Scanlon [14] and Broadbent [15]. Of these three algorithms, only Aspelund’s has been commercialized.

In parallel, our research group has developed a DR prediction algorithm, the first version of which was based on nine risk factors [16]. In the current version, we have included the presence or absence of DR as a tenth variable.

The aim of the present study is to demonstrate the effectiveness of our prediction algorithm based on ten variables in predicting the development of any type of DR or, in the case of mild DR, the risk of developing more advanced forms of DR, depending on the variables included in the algorithm. In this way, we can help treat patients with a high risk of DR more intensively and extend the screening time for those at a lower risk to between 12 and 36 months.

## 2. Materials and Methods

### 2.1. Setting

Hospital Universitario sant Joan de Reus, Tarragona, Spain.

### 2.2. Sample Size

We included retrospective data from the electronic health records (EHRs) of 120,382 T2DM patients, whom we followed over 11 years from 2010 to 2021 and for whom we had the necessary clinical data. The EHRs were provided by SIDIAP (“Sistema d’informació pel Desenvolupament de la Recerca a Atenció Primària”) [17]. The SIDIAP database contains a variety of patient data, such as those on visits to healthcare professionals, diagnostic codes, demographic information, clinical variables, laboratory tests results, prescriptions, referrals to specialists and hospitals, and medication dispensed in pharmacies. For this analysis, the data from an 11-year period (2010–2021) were extracted.

### 2.3. Inclusion Criteria

Patients with type 2 DM.Patients without DR or with mild DR.

### 2.4. Exclusion Criteria

Patients with type 1 DM.Patients included in diabetes group III and other specific types (i.e., diseases of the exocrine pancreas, endocrinopathy, genetic defects in ß-cell function, genetic defects in insulin action).Patients included in diabetes group IV and gestational diabetes mellitus (GDM).Patients who did not have a complete EHR.Patients with DR more serious than mild.

### 2.5. Construction of the Algorithm

The diabetic retinopathy prediction algorithm (DRPA) has been in development since 2010 for patients with T2DM. A total of 16 risk factors were considered: current age, age at diagnosis of DM, sex, type of DM, body mass index, duration of DM, DM treatment, smoking status, control of high blood pressure, diastolic pressure rate, systolic blood pressure rate, HbA1c%, creatinine levels, estimated glomerular filtration rate (eGFR) measured according to the CKD-EPI equation, total cholesterol level, LDL cholesterol level, HDL cholesterol level, triglyceride level, and microalbuminuria. Of these 16 risk factors in the first model, we included 9 in the first version of the algorithm [18].

Current ageSexBody mass indexDuration of T2DM in units of one yearT2DM treatment, diet, oral antidiabetics, insulin, insulin analoguesControl of arterial hypertension (normal values: systolic BP < 140, diastolic BP < 90)HbA1c% in 1% fractionsEstimated glomerular filtration rate, calculated from plasma creatinine using the chronic kidney disease epidemiology collaboration equation (CKD-EPI equation)Microalbuminuria value 30 mg/min up to 300 mg/min

In the version used in the present study, we included a tenth risk factor, which was the presence or absence of mild DR, in order to evaluate the risk of progressing from initial forms of DR to more severe ones.

10.Mild DR (yes or no)

In terms of how the algorithm was built, the training set information on the patients (on the ten selected variables) was used to automatically build a CDSS that computed the patients’ risk of developing DR. The system classified patients into one of two classes (with/without DR risk) and provided a numerical degree of certainty for the prediction. The CDSS was based on a fuzzy random forest (FRF). 

An FRF is a collection of fuzzy decision trees, in which each node corresponds to an attribute, each child of a node corresponds to a possible value of the variable, and each leaf of the tree corresponds to one of two possible classes (Figure 1). 

When a patient is classified by a fuzzy decision tree, the branch of the tree that corresponds to the values of the variables of the patient must be followed until the corresponding leaf is found and a prediction is made. Thus, each decision tree in the forest makes an individual prediction with a certain degree of certainty. The final prediction depends on the majority of the predictions of the single trees. 

The method used to develop the FRF was adapted from Yuan and Shaw [19]. The information on the patients is used to build a decision tree, iterating through the following steps: first, select randomly a small set of the remaining attributes, and second, check which attribute of that set discriminates better between the two classes. We made an empirical analysis using 10-fold cross-validation on the training set, considering the following ranges of values: 100, 200, or 300 trees; 1, 2, 3, or 4 randomly selected attributes in each node; and a leaf creation threshold between 0 and 1 (in 0.1 intervals). The best results on this validation were obtained using 200 trees (Figure 2; this shows that 200 trees is the point where the maximum values of accuracy, sensitivity, and specificity coincide), three selected attributes per node, and a high leaf creation threshold (0.8–1.0) [20]. The statistical results of the test set obtained an AUC value of 0.807%, a sensitivity of 80.67%, and a specificity of 85.96% [21].

### 2.6. Statistical Methods

The data were analyzed using SPSS version 22.0 (IBM® Statistics, Chicago, IL, USA). Descriptive statistical analysis of the quantitative data was carried out according to determination of the means and standard deviations. For the qualitative data, we used an analysis of the frequencies and percentages in each category. Univariate studies were carried out using two-sample Student’s *t*-tests to compare the quantitative variables, and we used a chi-squared table for the qualitative data. A *p* value of less than 0.05 was considered statistically significant.

We measured the screening performance of the study using a confusion matrix/contingency 2 × 2 table. Given a classified data set, there were four basic combinations of actual and assigned: correct positive assignments, or true positives [TP]; correct negative assignments, or true negatives [TN]; incorrect positive assignments, or false positives [FP]; and incorrect negative assignments, or false negatives [FN] [22].

The statistical evaluation of the data set included accuracy (A); sensitivity [S]; specificity [SP]; a positive predictive value, or precision [PPV]; harmonic mean (F1 score); a negative predictive value [NPV]; the positive false discovery rate, or type 1 errors [α]; the negative false discovery rate, or type 2 errors [β]; and the area under the curve, or diagnostic effectiveness, expressed as the proportion of correctly classified subjects.

Sensitivity or recall = TP/(TP + FN). This is the proportion of the population for whom the test is correct. Specificity = TN/(FP + TN). In clinical diagnosis, the values of sensitivity and specificity are considered good when they exceed 80%.

Positive predictive value or precision = TP/(TP + FP). This is a proportion of the population with a given test result for whom the test is correct. Negative predictive value = TN/(FN + TN).

The harmonic mean or F1 score = 2xprecisionxrecall/precision+recall, also known as the Sørensen–Dice coefficient or Dice similarity coefficient (DSC). In statistical analysis of binary classifications, the F1 score (also F-score or F-measure) is a measure of a test’s accuracy, calculated based on the positive predictive value and sensitivity of the test. It is a measure of the performance of the test that combines precision (the predictive positive value) and recall (sensitivity) into one number according to a choice of weighing, simply equal weighing. The highest possible value of F1 is 1, indicating perfect precision and recall, and the lowest possible value is 0 if either the precision or the recall is zero.

Also, we determined the accuracy, or diagnostic effectiveness = TP + TN/(TP + FP + TN + FN), which is expressed as the proportion of correctly classified subjects among all subjects. The accuracy is affected by the prevalence. With the same sensitivity and specificity, the diagnostic accuracy of a particular test increases as the disease prevalence decreases. On the other hand, the area under the curve (AUC) is not affected by prevalence and allows the performance of the classifier to be represented as a single value, in addition to allowing two or more different classifiers to be compared. Values of accuracy and AUCs of >0.8 show an almost perfect correlation, while values of 0.6–0.79 imply a significant correlation, 0.4–0.59 a moderate correlation, 0.2–0.39 a regular correlation, and 0–0.2 a poor correlation.

## 3. Results

### 3.1. Demographic Data

Table 1 demonstrates the demographic values of the sample size. The sample in the present study was 120,389 patients. 

Table 2 shows the patients’ DR statuses at the beginning and at the end of the study. A total of 111,172 patients with no DR and 9207 patients with mild DR were recruited. During the 11-year follow-up, 18,694 of those patients were diagnosed with DR (15.5%), of whom 5775 patients (4.77%) developed RDR (moderate DR or more) and 1581 patients (1.31%) developed sight-threatening DR (severe DR, proliferative DR, or diabetic macular edema).

### 3.2. Statistical Analysis of the Confusion Matrix/Contingency

Table 3 shows that the algorithm has both sensitivity and specificity within the limits considered necessary by the British Diabetes Association [16]: its sensitivity is above 80% both when predicting any type of DR and also when predicting RDR. Furthermore, from the metrics that determine whether the predicted cases are correctly classified, either positively or negatively, we can see that the precision (PPV) is 78% for any type of DR and 76% for RDR, which is a significant correlation. If we look at the harmonic mean (HM, or F1 score), which combines both precision and recall (sensitivity), the values are significant at 83% for any type of DR and 79% in the case of RDR. 

Regarding accuracy, it was identical in both cases, the value being 97% for both any type of DR and RDR. Accuracy is affected by the prevalence of the disease; thus, the prevalence of DR is 15.5%, which is somewhat low. In these cases, there tends to be a high accuracy, such as that in the present study. On the contrary, the AUC is not affected by prevalence. Our results, with an AUC of 93% for any type of DR and 90% for RDR, are strong.

## 4. Discussion

The objective of the present study was to assess whether the algorithm that we had previously developed, and into which we had incorporated the presence of mild DR as a variable, was useful for predicting the evolution of cases into more advanced forms of DR. In our case, it was. RDR includes moderate DR, severe DR, proliferative DR, and diabetic macular edema. All of these forms of disease affect eyesight, so making these predictions will facilitate early intervention. In patients with the mildest form (mild DR), adjusting certain clinical variables (HbA1c, kidney function, blood pressure) will slow down the disease progression to forms that might affect their vision.

We proposed two phases. The first was to identify those patients who were likely to develop DR but who did not have DR at the beginning of the study, which is why we left outpatients with mild DR in this first phase. The results are interesting: with an AUC of 93%, a harmonic mean of 83%, a precision of 78%, and a recall (sensitivity) of 88%, these figures indicate that the algorithm is useful in this context. More importantly, the specificity was 98%, and the negative predictive value was 99%, meaning that the algorithm predicts with sufficient certainty which patients, due to their clinical characteristics, will not develop DR. Linking both positive and negative predictors allows us to identify patients at low risk of developing DR and who would be safe to scan less frequently, thus freeing up resources to target more urgent cases.

The second phase focused on patients who had developed advanced forms of DR (RDR) during the study period. The results here were somewhat less promising than those previously mentioned, but some important conclusions can be drawn from them. In this second phase of the study, the AUC values were 90%, with a precision of 76%, a harmonic mean of 79%, and a recall of 82%. These values are lower than those for the detection of any type of DR, but they are significant enough to be useful for predicting the development of RDR based on the clinical data of the algorithm and the presence or absence of mild DR as an added risk variable. In this case, however, it is very important that both the specificity and the negative predictive value are 99%; when the algorithm predicts that a patient is at low risk of developing RDR, this is safe.

The two phases of the study, predicting the development of both any type of DR and RDR, meet the criteria defined by the British Diabetic Association, which propose that diabetic retinopathy screening programs should satisfy a minimum of 80% sensitivity and specificity, values that the present study has achieved [21].

In order to compare our results with those of other authors, we must take into account that two types of algorithms have been developed, with some of these already on the market: the first type based on retinal images and another based on clinical data. Our algorithm belongs to the second group.

Currently, diabetic retinopathy prediction studies can be divided into those that use retinal images to make the predictions, that is, the presence of microaneurysms and their quantities and locations are used to make the predictions. These studies are more current than those based on clinical data, such as the one we have presented, but they suffer from the problem of having to examine existing DR cases to understand whether or not they will evolve into more serious forms. Of these studies, the most important has been that carried out by Cunha Vaz, an author who has examined the turnover of microaneurysms located in the macular area to predict the evolution of DR since 2009. This author has proposed the existence of three phenotypes of MS patients, in whom DR evolves differently: a first phenotype A that develops DR as DM duration progresses, a second type B more susceptible to blood–retina barrier breakdown and that develops diabetic macular edema, and a third C, with a propensity toward the development of retinal ischemia. The same author has developed a system called RetMarkerDR, which has CE marking and is ready to be marketed [23,24,25]. 

In the second group of algorithms, those based on clinical data, we can refer only to three authors who have developed algorithms that are in use in practical clinics or that are in a very advanced stage of development, like ours. The three studies chosen are the only ones that have developed prediction algorithms based only on clinical data and with a real and extensive population database that have also been applied a posteriori in the populations of origin of the studies and have CE marking for commercial use in Europe. Table 4 describes the three studies, which are Aspelund [13], Scanlon [14], and Broadbent [15]. If we compare the AUC of the present study with the three selected studies, it is higher than those obtained by the other authors.

Table 4 shows that the most widely applied algorithm has been developed by Aspelund [13]. It was constructed from the diabetic retinopathy database managed at the Ophthalmology Department of Aarhus University Hospital in Denmark. The database was built based on the clinical data of 5199 patients over 20 years, thus allowing the algorithm to be tested prospectively. The model provides a recommended interval for follow-up fundus screening for the presence of retinopathy affecting vision (STDR) of between 6 and 60 months. The algorithm has been further tested in other countries, such as Spain, where Soto Pedre [10] studied 508 patients with T2DM. The results showed that 3.1% developed STDR before their subsequent screening visit, with the value of the area under the curve (AUC) being 0.74. In the Netherlands, Van der Heijden [26] tested the algorithm on 76 patients for a mean of 53 months and reported an AUC value of 0.83. Finally, in the UK, Lund [11], using a sample of 9690 DM patients followed for 2 years, reported that the algorithm predicted the onset of DR stages with AUC values of 0.833 for the T2DM patients.

The second algorithm that has published results is the model developed by Scanlon et al. in Gloucestershire, United Kingdom [14]. This algorithm was validated with 15,877 patients, obtaining an AUC of 0.77 for predicting the development of STDR.

Finally, the model developed by Broadbent [15], also known as the Liverpool Risk Calculation Engine, was built to detect the risk of developing STDR. The statistical analysis of this algorithm found an AUC value of 0.88 in the prediction of STDR, with a sensitivity value of 0.61 and a specificity of 0.93. We should highlight that this is the only report that provided sensitivity and specificity data. This algorithm was tested on a population of DM patients from Liverpool (UK) and was published in the form of an action protocol called the ISDR protocol [15].

Our algorithm is different in that it predicts RDR and any type of DR, while the other three predict STDR. Furthermore, compared with the other three algorithms, we use ten variables, as shown in Table 5, while Broadbent uses seven variables, Aspelund uses five, and Scanlon uses only three. Only our algorithm and that of Broadbent have taken into account the absence or presence of DR at the beginning of the study as a variable for predicting STDR, which suggests that using more variables might improve the robustness of the results, since Broadbent’s algorithm has an AUC of 0.88, similar to ours at 0.90, for predicting STDR.

One limitation of the present study is that it was retrospective, meaning that patients who did not have information on all the variables available were excluded, which can generate bias in the data collection. Likewise, we had to rely on data collected from electronic health records, meaning that we had to assume that the data had been collected correctly. To avoid this bias, the reliability of SIDIAP helped in proving the veracity of the results collected.

## 5. Conclusions

We have developed an algorithm that reliably predicts the development of any type of DR in patients with type 2 diabetes and the development of referable forms of DR with a high degree of certainty. Additionally, we can identify patients who will not develop DR by applying our algorithm, which facilitates the extension of the screening intervals for diabetic retinopathy with greater security. 

## Figures and Tables

**Figure 1 diagnostics-14-00833-f001:**
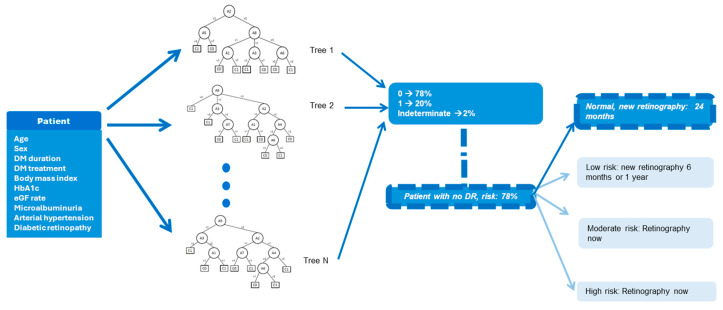
This figure shows how the data of the 10 variables are first entered into the algorithm. In the case represented, the diabetic retinopathy variable is equal to 0, that is, there is no previous retinopathy. After going through the algorithm, a result of a 78% probability of not having retinopathy is obtained, and from the four possible outputs (normal, low risk, moderate risk, and high risk) the normal value is chosen (strong blue color) and it is estimated that the next screening date may be in 24 months.

**Figure 2 diagnostics-14-00833-f002:**
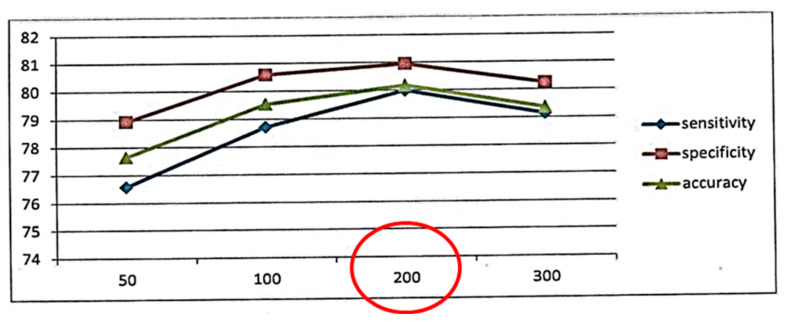
This figure shows that 200 trees is the point where the maximum value of accuracy, sensitivity, and specificity coincide.

**Table 1 diagnostics-14-00833-t001:** Demographic data of patients at the beginning of study.

	Mean
Age in years	68.01 ± 10.41
Men	68,578 (57%)
Women	51,811 (43%)
DM duration in years	9.11 ± 5.48
DM treatment: diet	11,840 (9.8%)
DM treatment: oral agents	92,325 (76.7%)
DM treatment: insulin	16,224 (13.5%)
Arterial hypertension	37,209 (30.9%)
Body mass index in kg/m^2^	27.86 ± 5.17
HbA1c in %	7.75 ± 1.59
Microalbuminuria mg/24 h	25.44 ± 125.92
CKD-EPI in mil/min/1.73 m^2^	75.08 ± 16.55

**Table 2 diagnostics-14-00833-t002:** Diabetic retinopathy classification at the beginning and end of the study.

	Patients’ Status at the Beginning of the Study	Percentage	Patients’ Status at the End of the Study	Percentage
No DR	111,172	92.36%	101,695	84.5%
Mild DR	9207	7.64%	12,919	10.7%
Moderate DR			4194	3.5%
Severe DR			598	0.5%
Proliferative DR			492	0.4%
Diabetic macular edema			491	0.4%
Total of patients with DR			18,694	15.5%

**Table 3 diagnostics-14-00833-t003:** Statistical confusion matrix of algorithm in our sample size. Note: for any type of DR, we have taken into account that 9477 patients developed some form of DR during the study and 5775 patients developed referable DR during the study.

	Any DR	RDR
True positive	8387	4727
False positive	2324	1466
True negative	108,588	113,148
False negative	1090	1048
Accuracy	0.97 (95% CI, 0.96–0.98)	0.97 (95% CI, 0.95–0.99)
AUC (area under the curve ROC)	0.93 (95% CI, 0.92–0.94)	0.90 (95% CI, 0.89–0.91)
Sensitivity or recall	0.88 (95% CI, 0.86–0.90)	0.82 (95% CI, 0.80–0.84)
Specificity	0.98 (95% CI, 0.96–0.99)	0.99 (95% CI, 0.95–0.994)
HM or F1 score	0.83 (95% CI, 0.81–0.84)	0.79 (95% CI, 0.78–0.80)
Precision or positive predictive values	0.78 (95% CI, 0.75–0.80)	0.76 (95% CI, 0.74–0.80)
Negative predictive values	0.99 (95% CI, 0.98–0.999)	0.99 (95% CI, 0.97–0.997)

**Table 4 diagnostics-14-00833-t004:** Comparison of the AUC curves between the different published algorithms.

Author (Name of Algorithm) Country	Country (Author) Type of Study	Number of Patients in Sample	AUC
Aspelund [13] (RETIRISK) Denmark	Denmark (Aspelund) Validation	5199 T1DM/T2DM patients with a 20-year follow-up	
	Spain (Soto Pedre) Real-world test	508 T1DM/T2DM patients	0.74
	Netherlands (van der Heijden) Real-world test	76 T1DM/T2DM patients with a 26-month follow-up	0.83
	United Kingdom (Lund) Validation	9690 T1DM/T2DM patient with a 2-year follow-up	0.83
Scanlon [14] United Kingdom	Gloucestershire (Scanlon) Real-world test	15,877 T1DM/T2DM patients	0.77
Broadbent [15] United Kingdom	Liverpool (Broadbent) Real-world test	4460 T1DM/T2DM patients	0.88
Romero-Aroca [16] (RETIPROGRAM) Spain	Spain (Romero-Aroca) Validation	101,802 T2DM patients	0.87
	Spain (Romero-Aroca) Real-world test	602 T2DM patients	0.98
	Spain (Romero-Aroca) Real-world test	120,384 T2DM patients with an 11-year follow-up Prediction of any type of DR	0.93
		120,384 T2DM patients with an 11-year follow-up Prediction of RDR	0.90

**Table 5 diagnostics-14-00833-t005:** Risk factors used by each algorithm.

	Aspelund	Scanlon	Broadbent	Authors
Current age	√	√		√
Age at diagnosis			√	
Sex			√	√
DM duration	√		√	√
DM treatment				√
Systolic blood pressure	√		√	√
Diastolic blood pressure				√
Total cholesterol		√	√	
HbA1c %	√	√	√	√
Microalbuminuria				√
Glomerular filtration rate measured using the CKD-EPI algorithm				√
Body mass index				√
DM type	√			
Diabetic retinopathy			√	√

## Data Availability

The original contributions presented in the study are included in the article, further inquiries can be directed to the corresponding authors.

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
