# Peer review of "Referable Diabetic Retinopathy Prediction Algorithm Applied to a Population of 120,389 Type 2 Diabetics over 11 Years Follow-Up"

_diagnostics, 2024, doi:10.3390/diagnostics14080833_

Round 1

Reviewer 1 Report

Comments and Suggestions for Authors

1. The literature survey / Background is missing in the manuscript. A lot of research on DR prediction has published. However, no such recent researches have been included in the manuscript. In fact, as I mentioned, the background with recent researches of DR prediction are missing in the manuscript. It must be there.

2. A lot of details is added on evaluation metrics. But the details of the algorithm used is very limited. It should be added. The algorithm model/block diagram / mathematical expressions may be be included.

3. Why the algorithm was built with 100 trees? why not less or more? Why not any other algorithm like support vector machine? What's the rationale for choosing this particular algorithm?

4. Three studies were chosen for comparisons in the research. Were they chosen only based on real world physically collected data of patients over a longer period? Please clear this.  

5. In abstract, you mentioned, "We validated 13 an algorithm for predicting DR and its progression by conducting a retrospective study of 11 children in a population of 120,389 T2DM". You mean 11 children or 11 years?

6. The details about the data is difficult to understand. It's better to add some graphical representations of statistical analysis, for better understanding of the reader.

Observation: It seems that the journal format has not been followed. The article style seems to be the one targeted for doctor's community, and presented in a way more like a medical study rather than a scientific study for general readers. 

Comments on the Quality of English Language

There are many spelling mistakes. The sentence structure also need to be reviewed.

Author Response

Revisions manuscript Title, Referable diabetic retinopathy prediction algorithm applied to a population of 120,389 type 2 diabetics over 11 years follow-up.

Reviewer 1

Comments and Suggestions for Authors

First, we would like to thank the reviewer for his dedication to this manuscript.

  1. The literature survey / Background is missing in the manuscript. A lot of research on DR prediction has published. However, no such recent researches have been included in the manuscript. In fact, as I mentioned, the background with recent researches of DR prediction are missing in the manuscript. It must be there.

Response.  Thank you for your comments, in the new version we have included new literature, including references to artificial intelligence articles that predict the development of Dr from initial forms such as those of Cunha VAz. We have explained more extensively in the introduction and discussion the two types of prediction algorithms, those based on images and those based on clinical data.sponse.

Question 2. A lot of details is added on evaluation metrics. But the details of the algorithm used is very limited. It should be added. The algorithm model/block diagram / mathematical expressions may be be included. And question 3. Why the algorithm was built with 100 trees? why not less or more? Why not any other algorithm like support vector machine? What's the rationale for choosing this particular algorithm?

Responses to questions 2 and 3

We included following sentence in methods.

How has the algorithm been built? The information of the patients for the training set (on the ten selected variables) was used to automatically build a CDSS that computes a patient’s patient’s risk of developing DR. The system classifies a patient into one of two classes (with/without DR risk) and provides a numerical degree of certainty in the prediction. The CDSS is based on a fuzzy random forest (FRF).

An FRF is a collection of fuzzy decision trees, in which each node corresponds to an attribute, each child of a node corresponds to a possible value of the variable, and each leaf of the tree corresponds to one of the two possible classes.

When a patient is classified by a fuzzy decision tree, the branch of the tree that corresponds to the values of the variables of the patient must be followed, until the corresponding leaf is found, and a prediction is made. Thus, each decision tree in the forest makes an individual prediction with a certain degree of certainty. The final prediction depends on the majority of predictions of the single trees. 

The method used to develop the FRF is adapted from Yuan and Shaw [15], the information about patients is used to build a decision tree, iterating the following steps, 1st select randomly a small set of the remaining attributes and 2nd check which attribute of that set discriminates better between the two classes. We made an empirical analysis using a 10-fold cross validation on the training set, considering the following ranges of values: 100-200-300 trees, 1-2-3-4 randomly selected attributes in each node, and a leaf creation threshold between 0 and 1 (in 0.1 intervals). The best results on this validation were obtained with 100 trees, three selected attributes per node, and a high leaf creation threshold (0.8–1.0). The statistical results of the test set obtained an AUC value of 0.807%, sensitivity of 80.67%, and specificity of 85.96% [16].

  1. Three studies were chosen for comparisons in the research. Were they chosen only based on real world physically collected data of patients over a longer period? Please clear this.  

Response

We included in text the paragraph in discussion

Currently, diabetic retinopathy prediction studies can be divided into those that use retinal images to make the prediction, that is, they are based on the presence of microaneurysms, their quantity and location to make the predictions. These studies are more current. than those based on clinical data like the one we present, but they suffer from having to already have diabetic retinopia to know whether or not it will evolve to more serious forms. Of these studies, the most important are those carried out by Cunha Vaz et al [ ] author who since 2009 has been working on the tournover of microaneurysms located in the macular area to predict the evolution of DR. This author has proposed the existence of three phenotypes of MS patients in which DR evolves dif-ferently, a first or phenotype A that develops DR with the duration of DM, a second or type B more susceptible to breaking the blood barrier retinal and who develop diabetic macualr edema and a third or C with a propensity to develop retinal ischemia. The same author has developed a system called RetMarkerDR that has CE marking to be marketed. In the second group of algorithms, those based on clinical data, we must refer only three authors who developed algorithms that are in use in the practical clinic or in a very advanced stage of being used like ours. The three studies chosen are the only ones that have developed pre-diction algorithms based only on clinical data and with a real and extensive population database, which have also been applied a posteriori in the populations of origin of the study and some have CE mark for commercial use in Europe. Table 4 describe the three studies, that are Aspleund [17], Scanlon [18] and Broadbent [19]. If we compare the AUC of the present study with the three se-lected studies it shows that is higher than those obtained by the other authors.

  1. In abstract, you mentioned, "We validated 13 an algorithm for predicting DR and its progression by conducting a retrospective study of 11 children in a population of 120,389 T2DM". You mean 11 children or 11 years?

Response In fact it is a bug that I have corrected in the new version, there are not 11 children are 11 years follow-up. Thank you very much for your appreciation.

  1. The details about the data is difficult to understand. It's better to add some graphical representations of statistical analysis, for better understanding of the reader.

Response. Thanks for your comments. I have rewritten many parts of the text, a new introduction, a better explained methods section and more concise results. I hope the new text is more readable.

Observation: It seems that the journal format has not been followed. The article style seems to be the one targeted for doctor's community, and presented in a way more like a medical study rather than a scientific study for general readers. 

Response Thank you very much for your comment. Indeed, being a doctor it is sometimes difficult to express myself in a way that is not exclusively to the medical world. In the new version we have tried to improve the writing to make it more understandable.

Comments on the Quality of English Language

There are many spelling mistakes. The sentence structure also need to be reviewed.

Response. Dear reviewer, we have sent the manuscript to the MDPI language reviewers for review.

Reviewer 2 Report

Comments and Suggestions for Authors

This study is relevant for the field and presented in a well-structured manner (include introduction, good described materials and methods section, separately described results, discussion and conclusions). The methods used in the present study are well described and justified by proper references. I would suggest that the manuscript can be considered for publication if the authors can improve the manuscript in following aspects:

One of the main limitations of the study, from the point of view of this reviewer, which should be pointed out in the introduction, is the aim is not expressed in an intelligible way according to hypothesis.

In the Background of the Abstract section, the question addressed is not placed in a broad context and the aim of the study is not emphasized. Please highlight the purpose of the study in the abstract section in accordance with the hypothesis.

Line 20: "Keyword" was written twice. It needs to be corrected.

Line 37-41: Reference 6 is not cited. Reference 6 is not mentioned at all in the manuscript.

Reference can be made for the formulas mentioned in the 2.7. Statistical Methods section of the manuscript.

Reference should be made for the statistical information mentioned.

Reference 28 is cited in the results section. Before reference 28, reference 15 was made in the method section. References 15 to 28 are not specified. The discussion section then continues with references 16, 17... etc. Please review all references and cite them in proper order. It should be checked that all references are cited in the correct place and should be cited in the same form and number in the references section. There are no more than 22 references. This confusion must be avoided.

The result section should be composed solely from result.  So please state the result, and nothing but the result. Any comments, reference cited and explanations should be omitted and moved to methods and discussion accordingly. 

In the references section, reference 21 is numbered twice.

There are no more than 22 references. This confusion must be avoided.

There is no reference 23. It is incorrect to specify a reference number as "Disclaimer/Publisher's Note:". These inaccuracies regarding references should be corrected.

Table 5 is not specified and explained in the manuscript.

The manuscript needs revision for language and grammar. There are spelling mistakes and word errors. It should definitely be reviewed and checked.

Comments on the Quality of English Language

The manuscript needs revision for language and grammar. There are spelling mistakes and word errors. It should definitely be reviewed and checked.

Author Response

Revisions manuscript Title, Referable diabetic retinopathy prediction algorithm applied to a population of 120389 type 2 diabetics over 11 years follow-up.

Reviewer 2

Comments and Suggestions for Authors

First, we would like to thank the reviewer for his dedication to this manuscript.

This study is relevant for the field and presented in a well-structured manner (include introduction, good described materials and methods section, separately described results, discussion and conclusions). The methods used in the present study are well described and justified by proper references. I would suggest that the manuscript can be considered for publication if the authors can improve the manuscript in following aspects:

  1. One of the main limitations of the study, from the point of view of this reviewer, which should be pointed out in the introduction, is the aim is not expressed in an intelligible way according to hypothesis.

Response. Thank you very much for your comment, in the new version we have rewritten the introduction section making it more extensive and I think we have improved the description of the aim of the study.

  1. In the Background of the Abstract section, the question addressed is not placed in a broad context and the aim of the study is not emphasized. Please highlight the purpose of the study in the abstract section in accordance with the hypothesis.

Response. We have changed the abstract completely, I hope it is more understandable.

  1. Line 20: "Keyword" was written twice. It needs to be corrected.

Response. In the new version we have solved the error

  1. Line 37-41: Reference 6 is not cited. Reference 6 is not mentioned at all in the manuscript.

Response. In the new version we have solved the error

  1. Reference can be made for the formulas mentioned in the 2.7. Statistical Methods section of the manuscript. Reference should be made for the statistical information mentioned.

Response. We have referenced the statistical methods section.

  1. Reference 28 is cited in the results section. Before reference 28, reference 15 was made in the method section. References 15 to 28 are not specified. The discussion section then continues with references 16, 17... etc. Please review all references and cite them in proper order. It should be checked that all references are cited in the correct place and should be cited in the same form and number in the references section. There are no more than 22 references. This confusion must be avoided.

Response. Dear reviewer, we have reviewed all the references in the writing and corrected the errors.

  1. The result section should be composed solely from result.  So please state the result, and nothing but the result. Any comments, reference cited and explanations should be omitted and moved to methods and discussion accordingly. 

Response. Dear reviewer, we have reviewed the following sections: introduction, we have made it new, methods, we have expanded it, results, we have eliminated the paragraphs that were specific to the discussion, we have expanded the discussion to DR image prediction systems.

  1. In the references section, reference 21 is numbered twice.

Dear reviewer, we have reviewed all the references in the writing and corrected the errors.

  1. There are no more than 22 references. This confusion must be avoided.

Dear reviewer, we have reviewed all the references in the writing and corrected the errors.

  1. There is no reference 23. It is incorrect to specify a reference number as "Disclaimer/Publisher's Note:". These inaccuracies regarding references should be corrected.

Dear reviewer, we have reviewed all the references in the writing and corrected the errors.

  1. Table 5 is not specified and explained in the manuscript.

Response. We included reference of table 5 in discussion text.

  1. Comments on the Quality of English Language

The manuscript needs revision for language and grammar. There are spelling mistakes and word errors. It should definitely be reviewed and checked.

Response. Dear reviewer, we have sent the manuscript to the MDPI language reviewers for review.

Reviewer 3 Report

Comments and Suggestions for Authors

After reviewing the article, I have identified several points that I believe are unreasonable:

1.The introductory sentence in the first paragraph lacks sufficient literature support. It states, "Diabetes mellitus (DM) is a chronic disease differentiated into two types: type 1 patients (T1DM) (young patients with an initial onset that requires treatment with insulin and possibly associated with the group of autoimmune diseases with a possible viral trigger) and type 2 patients (T2DM), which is the more frequent form and associated with insulin resistance problems secondary to obesity and metabolic syndrome."

2.In line 120, there is a mention of "Specificity= FP/(FP+ TN)", which should be corrected to "Specificity= TN/(TN+FP)" based on available information.

3.In line 123, "Negative predictive value= FN/(FN+ TN)" should be adjusted to "Negative predictive value= TN/(FN+ TN)" for accuracy.

4.The term "Predictive positive value" should be replaced with "Positive predictive value."

5.Additionally, it is noted that the abstract of the paper is too concise and lacks sufficient detail in its narrative.

Author Response

Revisions manuscript Title, Referable diabetic retinopathy prediction algorithm applied to a population of 120389 type 2 diabetics over 11 years follow-up.

Reviewer 3

Comments and Suggestions for Authors

First, we would like to thank the reviewer for his dedication to this manuscript.

After reviewing the article, I have identified several points that I believe are unreasonable:

1.The introductory sentence in the first paragraph lacks sufficient literature support. It states, "Diabetes mellitus (DM) is a chronic disease differentiated into two types: type 1 patients (T1DM) (young patients with an initial onset that requires treatment with insulin and possibly associated with the group of autoimmune diseases with a possible viral trigger) and type 2 patients (T2DM), which is the more frequent form and associated with insulin resistance problems secondary to obesity and metabolic syndrome."

Response. Dear reviewer, thank you very much for the time spent reading the text. Regarding the introduction, we have rewritten it in its entirety, eliminating paragraphs that could cause confusion.

2.In line 120, there is a mention of "Specificity= FP/(FP+ TN)", which should be corrected to "Specificity= TN/(TN+FP)" based on available information.

Response. We have corrected the errors in the sensitivity and specificity formulas.

3.In line 123, "Negative predictive value= FN/(FN+ TN)" should be adjusted to "Negative predictive value= TN/(FN+ TN)" for accuracy.

Response. We have corrected the errors in the sensitivity and specificity formulas.

4.The term "Predictive positive value" should be replaced with "Positive predictive value."

Response. We have substituted predictive positive value for positive predictive value.

5.Additionally, it is noted that the abstract of the paper is too concise and lacks sufficient detail in its narrative.

Response. We have rewritten the abstract, as well as the introduction, we have expanded methods and improved results and discussion

Round 2

Reviewer 1 Report

Comments and Suggestions for Authors

Thank you for addressing many of my concerns. However, I still think that illustrations are important to let readers understand about the data, the method and the results. Tables are Ok for results, but graphical representation of data and flow chart / block diagram of the algorithm are important. I would strongly recommend to add at least these two graphical details. I hope a computer scientist can understand and do that conveniently. It's essential for scientific readers. 

Comments on the Quality of English Language

English language improvement is required. It's good to know that the manuscript will be reviewed by the native language reviewers.

Author Response

Reviewer 1

First, I want to thank you for the time dedicated to reviewing the manuscript and for providing your interesting comments.

Commentary

Thank you for addressing many of my concerns. However, I still think that illustrations are important to let readers understand about the data, the method and the results. Tables are Ok for results, but graphical representation of data and flow chart / block diagram of the algorithm are important. I would strongly recommend adding at least these two graphical details. I hope a computer scientist can understand and do that conveniently. It's essential for scientific readers. 

Response

Many thanks for your comment, we have included two figures in the new version, one explaining how to calculate the risk of retinopathy in a patient and the second figure because 200 decision trees have been chosen.

On the other hand, I want to thank you for your insistence on this comment, since it has allowed me to realize an error in the previous version, the number of trees in the algorithm is 200 not 100, that is why I include figure 2 where I explain the reason for 200 trees and not 100 or 300. Thank you very much for this comment, I appreciate it

Reviewer 2 Report

Comments and Suggestions for Authors

I would like to thank the authors for providing a very detailed and comprehensive response to my comments from the previous review.

Author Response

Reviewer 2 second round

I would like to thank the authors for providing a very detailed and comprehensive response to my comments from the previous review.

Response

Thank you very much for the time dedicated to reviewing the manuscript and for providing your interesting comments.

Round 3

Reviewer 1 Report

Comments and Suggestions for Authors

Thank you for revision.